# Visual-Semantic Decomposition and Partial Alignment for Document-based Zero-Shot Learning

Xiangyan Qu
Institute of Information Engineering,
Chinese Academy of Sciences
School of Cyber Security University
of Chinese Academy of Sciences
Beijing, China
quxiangyan@iie.ac.cn

Jing Yu*
Institute of Information Engineering,
Chinese Academy of Sciences
School of Cyber Security, University
of Chinese Academy of Sciences
Beijing, China
yujing02@iie.ac.cn

Keke Gai
School of Cyberspace Science and
Technology,
Beijing Institute of Technology
Beijing, China
gaikeke@bit.edu.cn

Jiamin Zhuang
Institute of Information Engineering,
Chinese Academy of Sciences
School of Cyber Security, University
of Chinese Academy of Sciences
Beijing, China
zhuangjiamin@iie.ac.cn

Yuanmin Tang
Institute of Information Engineering,
Chinese Academy of Sciences
School of Cyber Security, University
of Chinese Academy of Sciences
Beijing, China
tangyuanmin@iie.ac.cn

Gang Xiong
Institute of Information Engineering,
Chinese Academy of Sciences
School of Cyber Security, University
of Chinese Academy of Sciences
Beijing, China
xionggang@iie.ac.cn

Gaopeng Gou
Institute of Information Engineering,
Chinese Academy of Sciences
School of Cyber Security, University
of Chinese Academy of Sciences
Beijing, China
gougaopeng@iie.ac.cn

Qi Wu
Australia Institute of Machine
Learning,
University of Adelaide
Adelaide, Australia
qi.wu01@adelaide.edu.au

## Abstract

Recent work shows that documents from encyclopedias serve as helpful auxiliary information for zero-shot learning. Existing methods align the entire semantics of a document with corresponding images to transfer knowledge. However, they disregard that semantic information is not equivalent between them, resulting in a suboptimal alignment. In this work, we propose a novel network to extract multi-view semantic concepts from documents and images and align the matching rather than entire concepts. Specifically, we propose a semantic decomposition module to generate multi-view semantic embeddings from visual and textual sides, providing the basic concepts for partial alignment. To alleviate the issue of information redundancy among embeddings, we propose the local-to-semantic variance loss to capture distinct local details and multiple semantic diversity loss to enforce orthogonality among embeddings. Subsequently, two losses are introduced to partially align visual-semantic embedding pairs according to their semantic relevance at the view and word-to-patch levels. Consequently, we consistently outperform state-of-the-art methods under two

*Corresponding authors

*MM '24, October 28-November 1, 2024, Melbourne, VIC, Australia.*
© 2024 Copyright held by the owner/author(s).
ACM ISBN 979-8-4007-0686-8/24/10
https://doi.org/10.1145/3664647.3680829

document sources in three standard benchmarks for document-based zero-shot learning. Qualitatively, we show that our model learns the interpretable partial association. Code is available at https://github.com/MorningStarOvO/EmDepart.

## CCS Concepts

• **Computing methodologies** → *Computer vision*.

## Keywords

document-based zero-shot learning; visual-semantic decomposition; partial semantic alignment

### ACM Reference Format:

Xiangyan Qu, Jing Yu, Keke Gai, Jiamin Zhuang, Yuanmin Tang, Gang Xiong, Gaopeng Gou, and Qi Wu. 2024. Visual-Semantic Decomposition and Partial Alignment for Document-based Zero-Shot Learning. In *Proceedings of the 32nd ACM International Conference on Multimedia (MM '24), October 28-November 1, 2024, Melbourne, VIC, Australia.* ACM, New York, NY, USA, 10 pages. https://doi.org/10.1145/3664647.3680829

## 1 Introduction

Image recognition tasks have achieved significant success relying on enormous manually labeled data. However, it is impractical to collect and annotate all kinds of images. Zero-Shot Learning (ZSL) [31, 41] emerges as a promising paradigm to address this issue. ZSL aims to identify unseen classes by training on a set of seen classes. The key challenge in ZSL is how to leverage auxiliary information to transfer knowledge from seen to unseen classes.

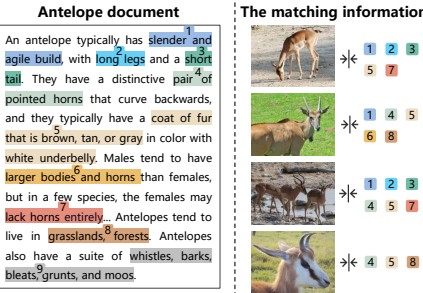

**Figure 1: Partial associations between documents and images. The semantic content in the category document may partially be reflected in the image. Distinct images capture varying aspects of the semantic information within the document.**

Common auxiliary information includes attributes [27, 53], word embeddings [24, 36, 42, 64], and category documents [6, 17, 45]. Most work [11, 13, 22, 30, 34] leverages human-annotated attributes as auxiliary information. These methods assume that seen and unseen classes can be defined with the same attributes. However, these attributes are labor-intensive and challenging to scale [47, 65], which are impossible for many real-world scenarios. To address this issue, other work [3, 21, 46, 63] applies category word embeddings from the pre-trained language model to replace attributes. However, the category name offers limited discriminative information [3, 8], imposing potential limitations on performance. Recently, some work [28, 37, 38] demonstrates that documents from the encyclopedias serve as valuable sources of auxiliary information, which contain multiple semantic concepts and knowledge from experts.

For document-based ZSL, seen and unseen classes are described by a composition of similar semantic concepts (noun phrases that visually describe a class). Aligning basic semantic concepts with corresponding image regions accurately is the key to knowledge transfer. Recent methods [37, 38] apply fine-grained interactions between words (or documents) and image patches to enhance semantic alignment. However, they are designed without considering the ***partial association*** between noisy documents and visual-diverse images: **1) Noisy document**: Documents from encyclopedias mainly cover many views, *e.g.* shape, color, habitat, sound, and diet. However, some views may not include visual information (see "9" in the left of Figure 1), *e.g.* sound and diet. These non-visual views are harmful to knowledge transfer. **2) Exhaustive description**: Documents comprehensively describe the possible characteristics of the category. However, a single image typically captures part of them. For example, the last image on the right of Figure 1 only shows the shape of the horn, color, and habitat in the antelope document. However, these methods align the entire semantics of documents with images, obtaining a suboptimal alignment. **3) Visual-diverse image content**: Due to variations in shooting angles, lighting, locations, and states, images of the same category convey varying semantic concepts from the document (see the right of Figure 1). Aligning diverse images with the same document semantics makes it hard to build accurate semantic alignment. Therefore, accurately modeling the partial association between document and image becomes an urgent problem for document-based ZSL.

To this end, we propose an **Em**bedding **De**composition and **Part**ial Alignment (EmDepart) network, as illustrated in Figure 2(b),

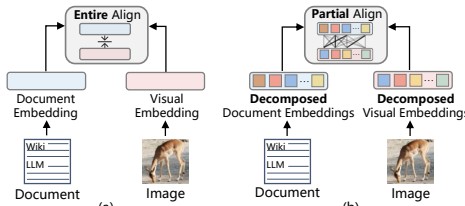

**Figure 2: Illustration of different methods. (a) Existing methods align the entire semantics of documents with images. (b) Our model decomposes semantic concepts and models the partial association to align the matching concepts accurately.**

to extract multi-view semantic concepts from document and image and accurately align the matching concepts. Specifically, the Semantic Decomposition Module (SDM) is proposed to generate multi-view semantic embeddings from visual and textual sides, providing the basic concepts for partial alignment. However, the SDM may generate multiple embeddings with a slight variance, resulting in information redundancy, denoted as feature collapse. To alleviate this issue, we propose the local-to-semantic variance loss to capture unique local details and multiple semantic diversity loss to make each embedding orthogonal to others. Subsequently, we rely on the semantic similarity of every visual and textual view embedding pair to model the partial association. Two losses are introduced to partially align these pairs according to their semantic relevance at the view and word-to-patch levels. Moreover, a novel score is applied to filter out unmatched information to measure semantic similarity accurately at the inference. Since some fine-grained categories are less described in the encyclopedia, we also design a novel prompt strategy to enrich these documents.

Our key contributions are as follows. (1) We propose a novel network that decomposes concepts from document and image into multi-view semantic embeddings and aligns them partially according to semantic relevance. This addresses the suboptimal alignment caused by ignoring the partial association in document-based ZSL. It sheds new light on the vision-and-language partial semantic alignment. (2) To solve the issue of information redundancy caused by feature collapse, we introduce the semantic decomposition module with the local-to-semantic variance loss to capture unique local details and multiple semantic diversity loss to enhance orthogonality among the embeddings. These losses also improve the performance of previous methods by 4.1% on average. (3) With comparable training parameters, our model consistently outperforms state-of-the-art methods for document-based ZSL and GZSL settings in three standard benchmarks. It improves performance by 6.0% and 5.8% on average across all metrics under Wiki and Wiki+LLM documents. Moreover, we qualitatively demonstrate that our model learns the interpretable partial semantic association.

## 2 Related work

**Zero-Shot Learning** aims to train on seen classes and generalize to recognize unseen classes [31, 41]. Most work leverages human-annotated attributes [15, 27, 32, 53] as auxiliary information. These methods transfer knowledge by utilizing compatibility functions [1, 2, 9, 44, 58] to map embeddings into a common space, incorporating generative model to generate unseen classes samples [52, 60, 61, 69], and enhancing semantic alignment between

attributes and image regions [11, 12, 22, 30, 34, 50, 62, 66, 67]. However, annotating attributes needs large human resources and deep domain expertise. In contrast, other work [3, 21, 46, 63] applies word embedding from pre-trained language models [24, 36, 42, 64], which transfers knowledge through the semantic relationship between different categories. Several methods [25, 26, 39, 54] enhance semantic connections by knowledge graphs. However, they achieve poor performance because of the category name with little discriminative information and sensitivity to linguistic issues [3, 8]. In contrast, documents are easy to collect from encyclopedias, which contain multiple semantic concepts. In this work, we improve knowledge transfer for document-based ZSL by accurate partial alignment.

**Document-based Zero-Shot Learning** uses definition-level text corpora from encyclopedias to obtain auxiliary information. Most work [4, 5, 8, 28, 43] utilize document embedding by TF-IDF [45] or large language models [6, 17] as auxiliary information. While several methods [19, 68] enhance embeddings by part detection network, annotated attributes are used to train the detection model. Recently, some work [37, 38] learns fine-grained interactions to enrich semantic embedding. Specifically, I2DFromer [38] trains a model to align image patches with words in global and local compatibility. I2MVFormer [37] aggregates information at the document level to reduce computation cost, aligning with image regions. However, these methods align the entire semantics of documents with images, ignoring the partial association between them. This results in suboptimal semantic alignment. In this work, our EmDepart generates multi-view semantic embeddings and models the partial association to align the matching semantics accurately.

**Set-based Embedding Methods** aims to learn multiple embeddings to alleviate the semantic ambiguity in cross-modal retrieval task, *i.e.*, an image is semantically matched with multiple captions. This is similar to the challenge for document-based ZSL, *i.e.*, the partial association between the document and diverse images. To be specific, PVSE [49] and TVMM [33] learn a set of embeddings by linear combination with local and global features. PCME [14] represents each sample as a probabilistic embedding. The state-of-the-art method [29] explores slot attention [35] to enhance diversity in set-based embeddings and smooth chamfer similarity to solve sparse supervision and set collapsing problems. However, text corpora in document-based ZSL are at definition level ($\approx 500$ words). It is hard to obtain multiple embeddings with little information redundancy solely through model architecture. Therefore, we propose two losses to enhance the information difference among semantic embeddings, alleviating the problem of feature collapse.

## 3 Method

Our **Em**bedding **De**composition and **Part**ial Alignment (EmDepart) network is illustrated in Figure 3. We first collect documents from encyclopedias and enrich less-described categories by LLMs. The image perceiver and text perceiver extract salient features for ZSL tasks. Then, the visual and textual Semantic Decomposition Modules (SDM) decompose perceived features into multi-view semantic embeddings. We leverage these embeddings to partially align the semantic information at the view and word-to-patch levels.

**Notations.** Zero-Shot Learning (ZSL) aims to train a classifier on seen classes $\mathcal{Y}^s$ to recognize unseen classes $\mathcal{Y}^u$ during the test,

where $\mathcal{Y}^s \cap \mathcal{Y}^u = \emptyset$. The training set $\{(x, y, d)|x \in \mathcal{X}^s, y \in \mathcal{Y}^s, d \in \mathcal{D}^s\}$ consists of image $x$, its label $y$ and auxiliary information, *i.e.*, document $d$. These documents are from a collection of textual descriptions of seen classes. At test time, another collection of images $\mathcal{X}^{test}$, their potential classes $\mathcal{Y}^{test}$, and corresponding documents $\mathcal{D}^{test}$ will be available to evaluate the model. In the ZSL setting, test images are from unseen classes, and for generalized ZSL (GZSL), from seen and unseen classes.

### 3.1 Document Collection

Category documents are the theoretical foundation for knowledge transfer in document-based ZSL. Each class (both seen and unseen classes) has a corresponding document that visually describes it.

**Document Collection from Encyclopedia.** Similar to [37, 38], we leverage the A-Z animals [56] for AWA2 [59], AllAboutBirds [57] for CUB [53], and Wikipedia [55] for FLO [40] to collect documents. Following previous methods, we select relevant sections in the encyclopedia to filter potential noises for the ZSL task.

**Enriching Less-Described Document.** Some fine-grained categories are less described in the encyclopedia, such as dogs chihuahua and collie in AWA2, Nighthawk and Green Violetear in CUB, and most classes in FLO. Therefore, we instruct Large Language Models (LLMs) to generate category definitions to enrich these documents by the following prompt:

*"Now you are a {type} expert. I will give you {type} name, and you need to give detailed visual information about its shape, color, appearance, habitat, etc. I want you to define {**class name**}."*

We use category species as {type}, *i.e.*, animal for AWA2, bird for CUB, and flower for FLO. We concatenate documents from encyclopedias with LLM-generated to serve as the final auxiliary information. To save computation costs, we enrich less-described instead of all categories. More details are shown in the supplementary.

### 3.2 Feature Extractor

**Image Perceiver.** Given an input image $x$, the image perceiver first encodes features by a fixed ViT [18]. Then, a learnable MLP layer with a residual connection maps image features to dimension $r$, extracting crucial visual information for ZSL tasks. The image perceiver outputs $[CLS]$ token $I_{CLS} \in \mathbb{R}^r$ as the global image feature and other tokens $I_l \in \mathbb{R}^{n \times r}$ as patch-wise local image features, where $n$ is the number of image patches.

**Text Perceiver.** Given a $m$-words document $d$, we use GloVe [42] to initiate each word as input features. Similar to previous work [37, 38], the text perceiver passes these token features through a learnable MLP with dimension $r$ ($r < 300$) to reduce computation cost. We add a learnable $[CLS]$ token to the sequence and input this sequence into the text encoder, which consists of two transformer encoder blocks. The text encoder outputs $[CLS]$ token $T_{CLS} \in \mathbb{R}^r$ as global text feature and $T_l \in \mathbb{R}^{m \times r}$ as word-wise local text features.

### 3.3 Semantic Decomposition Module

The semantics in images and documents are from multiple views, such as shape, color, and habitat. We introduce visual and textual semantic decomposition modules (SDM) to aggregate information from the perceived features of each modality and decompose them

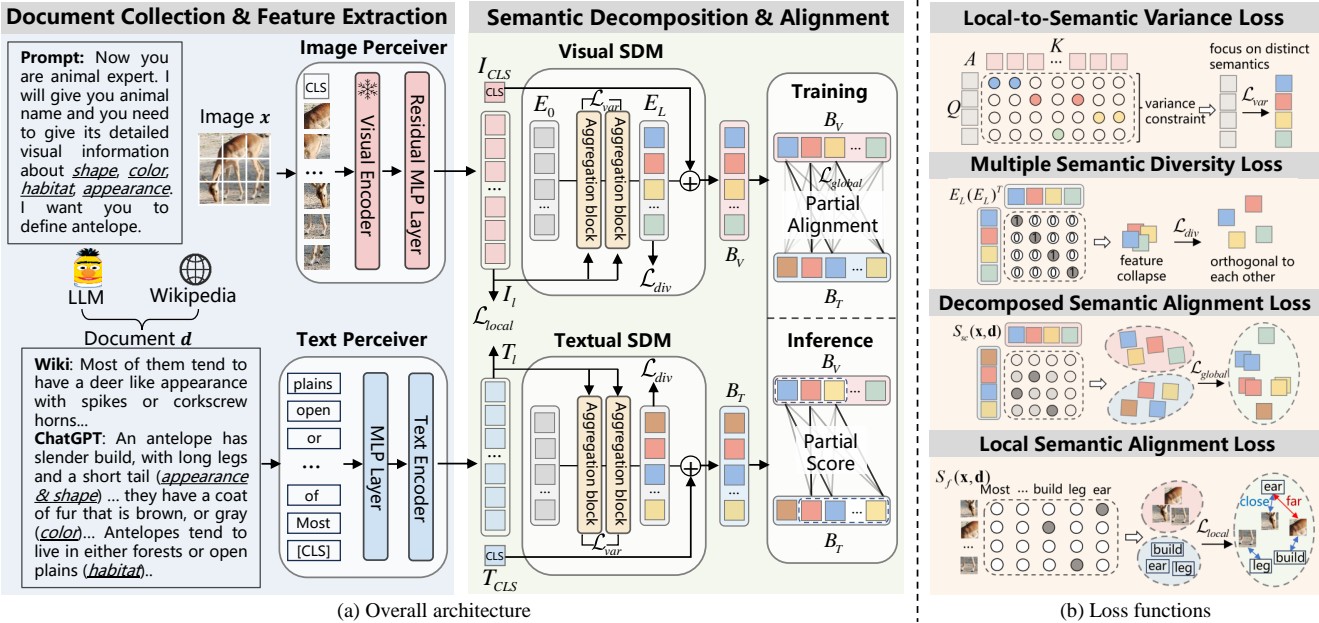

(a) Overall architecture       (b) Loss functions

**Figure 3: An overview of our model. (a) The EmDepart contains an image perceiver, a text perceiver, and visual and textual semantic decomposition modules. (b) Our loss functions. The first loss encourages each view embedding to focus on distinct local details. The second loss penalizes each embedding orthogonal to others. The last two losses partially align semantics at the view and word-to-patch levels.**

to generate multi-view semantic embeddings. This process provides the basic semantic concepts for partial alignment.

Taking visual SDM as an example, the textual side follows a similar process. As shown in the middle of Figure 3, SDM contains $l$-layer aggregation blocks integrating perceived features through iterations. In the initial iteration, we introduce a set of learnable tokens $E_0 \in \mathbb{R}^{k \times r}$, referred to as view embeddings later, where $k$ denotes the number of embeddings ($k \ll n$). Subsequently, for the $t$-th iteration, we feed both $E_{t-1}$ and local image features $I_l$ to the $t$-th aggregation block ($t = 1, 2, ..., l$), iteratively refining the visually discriminative information.

Each aggregation block aims to aggregate semantics and extract helpful visual information for the ZSL task. For the $t$-th aggregation block, we map local image features $I_l$ to key $K$ and value $V$ and view embeddings $E_{t-1}$ to query $Q$ by three linear layers. Then, they are fed into an attention mechanism to obtain the $\hat{E}_t \in \mathbb{R}^{k \times r}$:

$$\hat{E}_t = \text{softmax}\left(\frac{QK^T}{\sqrt{r_h}}\right)VW_o + E^{t-1}, \quad (1)$$

where $r_h$ is the dimension of head attention and $W_o \in \mathbb{R}^{r_h \times r}$ is a linear layer to map features to the original dimension $r$. Subsequently, we feed $\hat{E}_t$ to a learnable MLP followed by layer normalization, residual connection, and GELU activation [23], outputting the iterative visual-semantic information refinement of the view embeddings $E_t$:

$$E_t = \text{MLP}(\hat{E}_t) + \hat{E}_t. \quad (2)$$

In the last iteration, we concatenate the $E_L \in \mathbb{R}^{k \times r}$, the output of the final aggregation block, with $k$ repetitions of global image feature $I_{CLS}$ followed by a layer normalization. This operation constraints view embeddings with small within-set variance and

outputs the multi-view visual semantic embeddings $B_V \in \mathbb{R}^{k \times r}$:

$$B_V = \text{LayerNorm}(E_L + [I_{CLS}]^{\times k}). \quad (3)$$

Similarly, we obtain textual semantic embeddings $B_T \in \mathbb{R}^{k \times r}$.

### 3.4 Distinct Semantic Information Learning

Since definition-level corpora contain numerous words ($\approx 500$), view embeddings are hard to attend diverse semantics only by model architecture. The challenge also appears on the visual side. Specifically, the SDM may generate view embeddings with a slight variance, resulting in information redundancy, denoted as feature collapse. To solve this issue, we introduce two losses in SDM.

**Local-to-Semantic Variance Loss** aims to encourage each view embedding to focus on unique local information (see the first loss in Figure 3 (b)). Taking the visual side as an example, it enforces different view embeddings to show distinct attention to the same image patch. We make the following variance constraints on attention maps $A_V$ in visual SDM, a dot product between $Q$ and $K$:

$$C(A_V) = \sum_{t=1}^{l} \sum_{j=1}^{n} \max(0, \gamma - \sqrt{Var(a_{tj}) + \epsilon}), \quad (4)$$

where $a_{tj}$ denotes the attention between visual view embeddings and the $j$-th image patch in the $t$-th aggregation block. The $l$ and $n$ are the number of aggregation blocks and image patches, respectively. We offer a constant value $\gamma$ for constraints, which ensures view embeddings with a certain diversity to the attention of the same patch. The $\epsilon$ is a small scalar to maintain numerical stabilities. Similarly, we penalize attention maps $A_T$ between textual view

embeddings and each word token. This loss is formulated as:

$$\mathcal{L}_{var} = \frac{1}{2}\left(C(A_T) + C(A_V)\right). \tag{5}$$

After $l$ iterations, each visual and textual view embedding carries distinct semantic information, establishing the foundation for decomposing information.

**Multiple Semantic Diversity Loss** aims to enhance the information decoupling among view embeddings. It forces minimal semantic redundancy between view embeddings by making each embedding orthogonal to others, shown in the second loss of Figure 3(b). Notably, view embeddings of the final output ($B_V$ and $B_T$) contain the global feature, which may invalidate the orthogonality constraint. Therefore, we penalize redundancy among the output of the final aggregation block, i.e., $E_L$. Specifically, we normalize each $E_L$ and calculate the cosine similarity between them, yielding the redundant matrix $M_V = E_L(E_L)^T$. The loss penalizes the visual and textual redundant matrix (denoted as $M_T$) to approximate the identity matrix $\mathbb{I} \in \mathbb{R}^{k \times k}$ via an $l_2$-norm minimization:

$$\mathcal{L}_{div} = \frac{1}{2k^2}(\|M_T - \mathbb{I}\|_2 + \|M_V - \mathbb{I}\|_2), \tag{6}$$

where $M_T$ is computed in a similar way. This objective ensures that different view embeddings maintain orthogonality, characterized by non-diagonal cosine similarity values converging towards zero.

To summarize, since the $\mathcal{L}_{var}$ penalizes view embeddings to focus on different local information and the $\mathcal{L}_{div}$ constrains each embedding to be orthogonal to others, the SDM generates decomposed view embeddings with distinct semantic information.

## 3.5 Partial Semantic Alignment

To accurately model the partial association between documents and images, we introduce two losses to align the matching semantic concepts at the view and word-to-patch levels.

**Decomposed Semantic Alignment** aims to align decomposed visual and textual view embeddings according to their semantic relevance. We rely on the semantic similarity between visual and textual view embedding pairs to model the partial association by the Smooth Chamfer [29] function. We first review the Smooth Chamfer, which assigns distinct weights to every document-image embedding pair based on similarity:

$$S_{sc}(x, d) = \frac{1}{2k}\left(\sum_{b_T \in B_T} \text{LSE}(b_T, B_V) + \sum_{b_V \in B_V} \text{LSE}(b_V, B_T)\right), \tag{7}$$

where $\text{LSE}(b, B)$ is smooth approximation for maximum cosine similarity between vector $b$ and elements in set $B$. Taking $\text{LSE}(b_T, B_V)$ as an example, it is formulated as:

$$\text{LSE}(b_T, B_V) = \log\left(\sum_{b_V \in B_V} e^{\cos(b_T, b_V)}\right), \tag{8}$$

where $\cos(\cdot)$ is the cosine similarity. We introduce a cross-entropy loss to encourage image $x$ and corresponding document $d$ to be closer than other pairs:

$$\mathcal{L}_{global} = -\log \frac{\exp(S_{sc}(x, d)/\tau)}{\sum_{d' \in \mathcal{D}^s} \exp(S_{sc}(x, d')/\tau)}, \tag{9}$$

where $\tau$ is a temperature scalar. The $\mathcal{L}_{global}$ is designed to smoothly align each visual embedding in $B_V$ with the most similar element

from textual embeddings $B_T$, and vice versa (see the third loss in Figure 3(b)). This process models the partial association between visual and textual spaces, which embodies the fact that an image is reflected as part of semantics in the document. Consequently, we align the two spaces more accurately.

**Local Semantic Alignment** aims to apply interactions between image patches and word tokens for fine-grained semantic alignment (see the fourth loss in Figure 3(b)). It provides the basic semantic concepts for partial semantic alignment and discriminative information for fine-grained classification. Similar to [37, 38], we first fuse local image and text features by a cross-attention module, which leverages the semantic information in the document to enrich the visual features. The cross-attention module takes local image features $I_l$ as query and local text features $T_l$ as key and value, outputting semantic-enhanced visual features $\tilde{I} \in \mathbb{R}^{n \times r}$. Subsequently, we apply a global pooling on patch dimension to aggregate the visually fine-grained information, yielding the $\bar{I} \in \mathbb{R}^{1 \times r}$. Afterward, a fine-grained similarity score $S_f(x, d) = \text{D}(\bar{I})$ is introduced through a linear layer $\text{D}(\cdot) \in \mathbb{R}^{r \times 1}$. The objective is to encourage the image $x$ to be close to the corresponding category document $d$ on fine-grained score, optimizing with a cross-entropy loss:

$$\mathcal{L}_{local} = -\log \frac{\exp\left(S_f(x, d)\right)}{\sum_{d' \in \mathcal{D}^s} \exp\left(S_f(x, d')\right)}. \tag{10}$$

**Training.** Our EmDepart is optimized with the following loss:

$$\mathcal{L} = \mathcal{L}_{global} + \lambda_{local}\mathcal{L}_{local} + \lambda_{var}\mathcal{L}_{var} + \lambda_{div}\mathcal{L}_{div}, \tag{11}$$

where $\lambda_{local}$, $\lambda_{var}$, and $\lambda_{div}$ are hyper-parameters. The joint training enhances EmDepart to generate multi-view semantic embeddings with information decoupling and accurately align visual and textual space to a common semantic space according to the matching information, significantly improving knowledge transfer.

## 3.6 Inference

During the inference, a partial score is proposed to filter out unmatched information to measure semantic similarity accurately.

**Partial Score Function.** In Figure 1, it is evident that an image semantically matches a subset of the semantic concepts within the document. Similarly, we limit the similarity computation solely to document-image semantic pairs with the highest $p$ similarity values, where $p < k$. Specifically, we select the top $p$ similarity values between each visual view embedding and all textual view embeddings, yielding a total of $p \times k$ pairs. Subsequently, a similar process is applied to each textual view embedding, resulting in the selection of $p \times p$ pairs. We denote this process as $\text{TopCos}(x, d, p)$ function, effectively filtering out unmatched information between documents and images. The smooth chamfer is then used to compute the similarity among these chosen pairs:

$$S_p(x, d) = S_{sc}(\text{TopCos}(x, d, p)). \tag{12}$$

**Inference.** Given an input image $x$, we obtain a prediction $\hat{y}$ that yields the highest partial score $S_p(x, d')$ among unseen classes for ZSL, i.e., $\mathcal{D} = \mathcal{D}^u$, and among both seen and unseen classes for GZSL, i.e., $\mathcal{D} = \mathcal{D}^s \cup \mathcal{D}^u$:

$$\hat{y} = \arg\max_{d' \in \mathcal{D}} S_p(x, d'). \tag{13}$$

**Table 1: Comparison with SOTA methods in document-based ZSL. We evaluate methods on documents sourced from Wiki and Wiki+LLM. The best results are in bold. Performance gain compared to methods on the same document source is in blue.**

| Model | Auxiliary Information | Zero-Shot Learning | | | Generalized Zero-Shot Learning | | | | | | | | |
|---|---|---|---|---|---|---|---|---|---|---|---|---|---|
| | | AWA2 | CUB | FLO | AWA2 | | | CUB | | | FLO | | |
| | | T1 | T1 | T1 | U | S | H | U | S | H | U | S | H |
| GloVe [42] | CLSN | 52.1 | 20.4 | 21.6 | 42.1 | 75.3 | 54.0 | 16.2 | 43.6 | 23.6 | 14.4 | 88.3 | 24.8 |
| GloVe [42] | Wiki | 61.6 | 29.0 | 25.8 | 49.5 | 78.1 | 60.6 | 23.8 | 62.6 | 34.5 | 14.7 | 91.0 | 25.3 |
| LongFormer [6] | Wiki | 44.2 | 22.6 | 8.8 | 41.6 | 81.8 | 55.2 | 19.9 | 41.0 | 26.8 | 8.8 | 89.8 | 16.0 |
| MPNet [48] | Wiki | 61.8 | 25.8 | 26.3 | 58.0 | 76.4 | 66.0 | 20.6 | 44.3 | 28.2 | 22.2 | 96.7 | 36.1 |
| TF-IDF [45] | Wiki | 46.4 | 39.9 | 34.0 | 29.6 | 87.6 | 44.2 | 29.0 | 52.1 | 37.3 | 28.9 | 94.8 | 44.3 |
| VGSE [63] | CLSN+IMG | 69.6 | 37.1 | - | 56.9 | 82.8 | 67.4 | 27.6 | **70.6** | 39.7 | - | - | - |
| I2DFormer [38] | Wiki | 76.4 | 45.4 | 40.0 | 66.8 | 76.8 | 71.5 | 35.3 | 57.6 | 43.8 | 35.8 | 91.9 | 51.5 |
| I2MVFormer [37] | Wiki | 73.6 | 42.1 | 41.3 | 66.6 | 82.9 | 73.8 | 32.4 | 63.1 | 42.8 | 34.9 | 96.1 | 51.2 |
| **EmDepart** (Ours) | Wiki | **81.4**$^{+5.0}$ | **50.2**$^{+4.8}$ | **47.2**$^{+5.9}$ | **76.0** | **87.8** | **81.5**$^{+7.7}$ | **42.6** | 56.3 | **48.5**$^{+4.7}$ | **42.7** | **97.6** | **59.5**$^{+8.0}$ |
| I2DFormer [38] | Wiki+LLM | 77.3 | 47.0 | 43.0 | 68.6 | 77.4 | 72.7 | 38.5 | 59.3 | 46.7 | 40.4 | 80.1 | 53.8 |
| I2MVFormer [37] | Wiki+LLM | 79.6 | 51.1 | 46.2 | 75.7 | 79.6 | 77.6 | 42.5 | 59.9 | 49.7 | 41.6 | 91.0 | 57.1 |
| **EmDepart** (Ours) | Wiki+LLM | **86.1**$^{+6.5}$ | **52.8**$^{+1.7}$ | **53.3**$^{+7.1}$ | **81.4** | **88.5** | **84.8**$^{+7.2}$ | **45.0** | 61.4 | **51.9**$^{+2.2}$ | **52.3** | 94.4 | **67.3**$^{+10.2}$ |

**Table 2: Comparison with set-based embedding methods. Performance improvement after adding our losses is in blue.**

| Model | AWA2 | | CUB | | FLO | |
|---|---|---|---|---|---|---|
| | T1 | H | T1 | H | T1 | H |
| TVMM [33] | 77.4 | 74.4 | 41.6 | 43.1 | 42.3 | 54.2 |
| +$\mathcal{L}_{var}$ + $\mathcal{L}_{div}$ | 81.0 | 77.5 | 45.6 | 47.4 | 46.8 | 59.5 |
| Gain | +3.6 | +3.1 | +4.0 | +4.3 | +4.5 | +5.3 |
| S-Chamfer [29] | 81.5 | 80.6 | 45.6 | 45.2 | 43.5 | 57.3 |
| +$\mathcal{L}_{var}$ + $\mathcal{L}_{div}$ | 84.0 | 82.9 | 49.1 | 49.9 | 48.9 | 63.6 |
| Gain | +2.5 | +2.3 | +3.5 | +4.7 | +5.4 | +6.3 |
| **EmDepart**(Ours) | **86.1** | **84.8** | **52.8** | **51.9** | **53.3** | **67.3** |

## 4 Experiments

**Datasets.** We evaluate on three benchmark datasets, *i.e.*, a coarse-grained dataset Animals with Attributes2 (AWA2) [59], two fine-grained datasets Caltech-USCD Birds-200-2011 (CUB) [53] and Oxford Flowers (FLO) [40]. The seen-unseen class division is from Proposed Split [59]. We use documents instead of human-annotated attributes in datasets as auxiliary information.

**Evaluation Metrics.** Following [59], we measure the average per-class top-1 accuracy (T1) on unseen classes for ZSL. For GZSL, we present the per-class mean accuracy on seen ($S$) and unseen classes ($U$) as well as their harmonic mean $H = 2 \times U \times S/(U + S)$.

**Implementation Details.** Similar to [37, 38], we utilize the ViT-B/16 [18] pre-trained on ImageNet 1K [16] as the visual backbone. If not noted otherwise, we show the performance of ChatGPT [20] as LLMs. Hyperparameters are optimized by grid search in the validation split. Once the hyperparameters are confirmed, we merge the validation with the training split to obtain the performance on the test split. We also apply calibrated stacking [10] for GZSL to trade-off calibration degrees, reducing the bias towards seen classes. More details are available in the supplementary.

### 4.1 Comparing with the SOTA Methods

**Comparison with SOTA in Document-based ZSL.** In Table 1, we compare our EmDepart with state-of-the-art (SOTA) methods in document-based ZSL. For a fair comparison, we evaluate methods

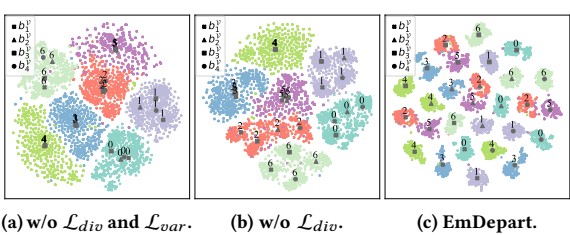

(a) w/o $\mathcal{L}_{div}$ and $\mathcal{L}_{var}$.    (b) w/o $\mathcal{L}_{div}$.    (c) EmDepart.

**Figure 4: Analysis of feature collapse. Each number denotes a class (same color), and each shape denotes one of the view embeddings. With the addition of $\mathcal{L}_{var}$ and $\mathcal{L}_{div}$, information differences between embeddings gradually increase.**

with the same text and image perceivers. EmDepart outperforms previous methods across all metrics (T1 and H) regarding ZSL and GZSL settings on all datasets. It confirms that modeling the partial association is beneficial for accurate semantic alignment. The previous SOTA methods [37, 38] align complete semantics in documents with images, thus hindering knowledge transfer.

**Wiki vs Wiki+LLM Documents.** With Wiki documents, EmDepart achieves optimal performance compared to previous methods. Notably, EmDepart with Wiki outperforms SOTA methods with Wiki and LLM regarding T1 and H on AWA2 and FLO. It confirms that modeling the partial association is significant for knowledge transfer. Besides, we see a performance improvement from Wiki to Wiki+ChatGPT. This is because visual descriptions generated by LLMs enrich semantic information in less-described classes.

**Comparison with SOTA in Set-based Embedding Methods.** In Table 2, we replace SDM with TVMM [33] and S-Chamfer [29], the SOTA set-based embedding methods. Since documents are at the category level, it is challenging to rely solely on model architecture to produce view embeddings with little redundant information. SDM achieves the optimal performance by incorporating $\mathcal{L}_{div}$ and $\mathcal{L}_{var}$ to enhance information difference among view embeddings. Similarly, TVMM and S-Chamfer improve performance after adding $\mathcal{L}_{div}$ and $\mathcal{L}_{var}$, facilitating differences between embeddings.

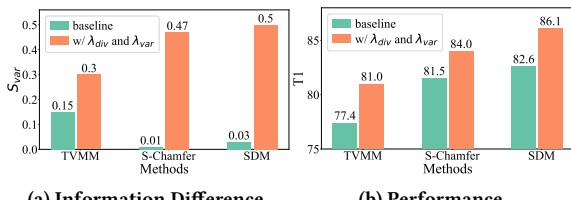

**(a) Information Difference.**  **(b) Performance.**

**Figure 5: Analysis of baseline and model after adding our losses. The larger $S_{var}$ denotes more distinct between embeddings, and $S_{var} = 0$ denotes embeddings are all the same.**

## 4.2 Analysis of Feature Collapse

In Figure 4(a), we show the feature collapse between view embeddings, *i.e.*, embeddings have little variance, resulting in information redundancy. It is harmful to model the partial association with the same embeddings. To solve this issue, we introduce $\mathcal{L}_{var}$ to make each embedding attend to distinct local details and $\mathcal{L}_{div}$ to penalize embeddings orthogonal to others. In Figure 4(a-c), view embeddings carry more distinct information (at more different positions) with the $\mathcal{L}_{var}$ and $\mathcal{L}_{div}$. Quantitatively, we introduce the circular variance $S_{var} = 1 - \|\sum_{b \in B} b/|B|\|_2$ to analyze the information difference of view embeddings. TVMM [33] and S-Chamfer [29] lead to feature collapse under category-level corpora in Figure 5. After adding $\mathcal{L}_{var}$ and $\mathcal{L}_{div}$, we improve these methods performance and increase the information difference among view embeddings.

## 4.3 Analysis of Partial Association

In Figure 6, we qualitatively show that our model learns the interpretable partial semantic association. It contains the visual-semantic decomposition to offer basic semantic concepts and partial semantic alignment according to the matching information.

**Visual-Semantic Decomposition.** We see that different view embeddings focus on distinct information in each modality. Utilizing the giraffe as an example, EmDepart focuses on appearance (in $b_V^1$ and $b_V^2$), habitat (in $b_V^4$), and global information (in $b_V^3$) for the visual side. Similarly, there are textual descriptions on color (in $b_T^1$), appearance and shape (in $b_T^2$ and $b_T^3$), and habitat (in $b_T^4$). This verifies that the SDM decomposes semantics from images and documents and generates multi-view semantic embeddings.

**Partial Semantic Alignment.** On the similarity matrix of $S_{sc}(x, d)$, we observe the accurate semantic alignment between document and image. In particular, since the second giraffe image does not represent the visual content about habitat and body, $(b_V^2, b_T^3)$ and $(b_V^4, b_T^4)$ has a high similarity in the first image but low in the second. In the first red tiger lily, $(b_V^2, b_T^2)$ has the highest score, while $(b_V^1, b_T^1)$ has the highest score in the second orange one. This is consistent with the fact that $b_T^1$ pays more attention to "orange" and $b_T^2$ focuses more on "red".

## 4.4 Ablation Study

We ablate key components of EmDepart in Table 3. For all models, we leverage documents from Wiki+LLM as auxiliary information.
**Ablation on Loss Functions.** In row b), we see a significant drop in performance on CUB and FLO. This is due to the lack of interaction between patches and words, which offers discriminative

**Table 3: Ablation of key components in EmDepart**

| Model | AWA2 | CUB | FLO |
|---|---|---|---|
| | T1 | T1 | T1 |
| a) full model | **86.1** | **52.8** | **53.3** |
| **Ablation on Loss Function** | | | |
| b) w/o $\mathcal{L}_{local}$ | 85.8 | 45.9 | 41.7 |
| c) w/o $\mathcal{L}_{div}$ | 83.5 | 47.7 | 41.5 |
| d) w/o $\mathcal{L}_{var}$ | 85.5 | 50.1 | 49.9 |
| e) w/o $\mathcal{L}_{div} + \mathcal{L}_{var}$ | 82.6 | 47.5 | 39.3 |
| f) w/o $\mathcal{L}_{local} + \mathcal{L}_{div} + \mathcal{L}_{var}$ | 80.1 | 45.4 | 37.2 |
| **Ablation on Score Function** | | | |
| g) w/o Partial Score in Eq.12 | 85.7 | 52.6 | 53.0 |
| h) w/ average distance in Eq.7 | 80.0 | 39.4 | 45.7 |
| i) w/ maximum distance in Eq.7 | 82.2 | 45.4 | 44.8 |
| **Ablation on Module** | | | |
| j) w/o global feature in Eq.3 | 71.6 | 37.7 | 39.6 |
| k) w/o SDM | 79.7 | 46.0 | 45.1 |
| l) w/o residual connection | 81.4 | 49.7 | 48.3 |

**Table 4: Ablation of LLMs. The error bars are obtained from three different documents generated by LLMs.**

| Auxiliary | AWA2 | | FLO | |
|---|---|---|---|---|
| Information | T1 | H | T1 | H |
| Wiki | 81.4 | 81.5 | 47.2 | 59.5 |
| Wiki+GPT3 [7] | $82.3_{\pm 0.45}$ | $82.2_{\pm 0.61}$ | $53.2_{\pm 0.78}$ | $65.5_{\pm 0.87}$ |
| Wiki+LLaMa2 [51] | $82.1_{\pm 0.37}$ | $82.8_{\pm 0.27}$ | $49.5_{\pm 0.65}$ | $62.8_{\pm 0.61}$ |
| Wiki+ChatGPT [20] | $\mathbf{86.1}_{\pm 0.16}$ | $\mathbf{84.8}_{\pm 0.29}$ | $\mathbf{53.3}_{\pm 0.41}$ | $\mathbf{67.3}_{\pm 0.82}$ |

information for fine-grained classification. The performance of removing $\mathcal{L}_{div}$ decreases more than $\mathcal{L}_{var}$ in rows c) and d). This is due to $\mathcal{L}_{var}$ constraining the variance in attention blocks, which means feature collapse may exist after the MLP projection in SDM. Row e) shows a further decrease in performance, which indicates the complementary of $\mathcal{L}_{div}$ and $\mathcal{L}_{var}$. Row f) achieves the worst performance, further verifying the effectiveness of our losses.

**Ablation on Score Functions.** The performance degrades in row g) due to the partial score filtering out unmatched semantic information, measuring similarity accurately in the inference. Rows h) and i) ablate Smooth Chamfer with the average distance in [14] and maximum distance in [49]. Smooth Chamfer performs better because it overcomes problems posed by sparse supervision in maximum distance and feature collapse in average distance.

**Ablation on Proposed Modules.** In row j), we remove the global feature in Eq.3. It performs worse as view embeddings using only local features may introduce a large variance, leading to overfitting on seen classes. In contrast, global features in Eq.3 ensure that the variance remains within a controlled range. In row k), we show the result of removing the SDM and leveraging the global feature to align entire semantics of documents to images like [37, 38]. The performance decreases due to the suboptimal semantic alignment, ignoring the partial association. Besides, performance drops in row l) when the visual perceiver lacks a residual connection, which preserves the original visual knowledge inherent to ViT [18], a crucial factor for knowledge transfer.

**Ablation of Different LLMs.** In Table 4, we show the effect of different LLMs, consistently improving the performance compared to Wiki documents. It verifies the effectiveness of enriching less-described documents. The performance in FLO improves significantly due to the lack of detailed descriptions for most classes in Wiki. The ChatGPT [20] achieves the best result, which generates more detailed descriptions with rich semantics.

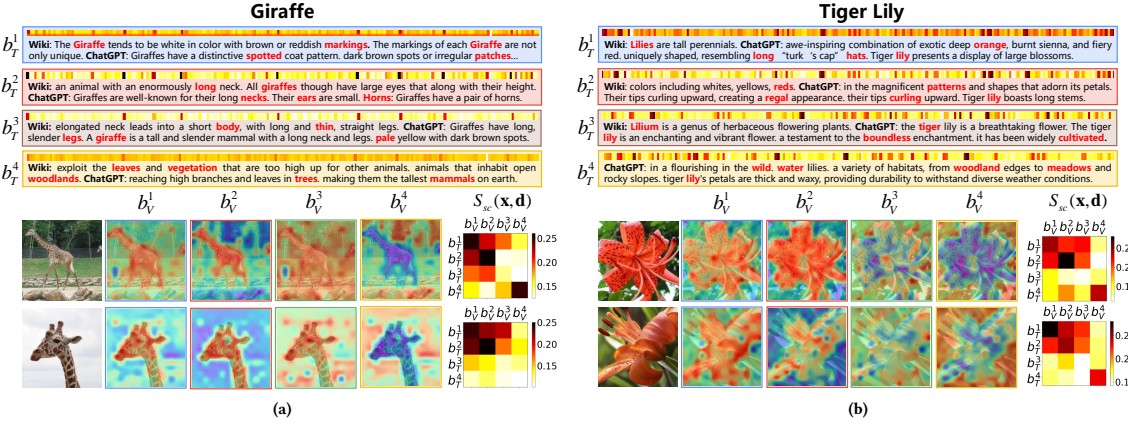

**Figure 6: Partial association analysis on AWA2 and FLO datasets. We present attention maps for each visual and textual view embedding, the top 5 most attended words (in red) with nearby words, and smooth chamfer score. In attention maps, darker colors represent larger attention values. Our EmDepart achieves accurate semantic alignment on the matching information.**

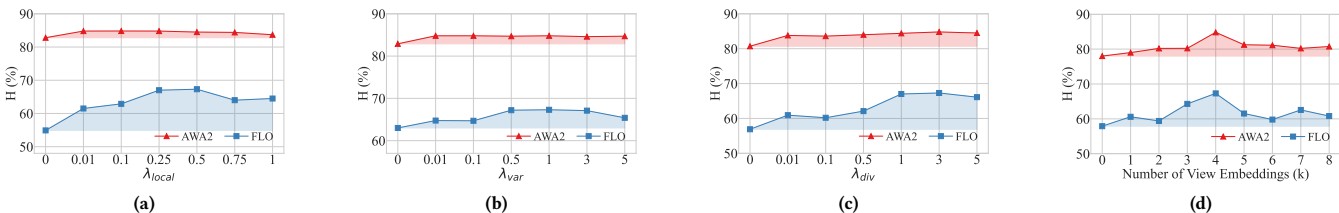

**Figure 7: Effect of loss weights (a-c) and number of view embeddings (d) on coarse-grained AWA2 and fine-grained FLO datasets. The shaded area indicates the performance improvement compared to hyperparameters set as 0.**

## 4.5 Impact of Hyperparameters

**Effect of Loss Weights.** In Figure 7(a-c), as the $\lambda_{local}$, $\lambda_{var}$, and $\lambda_{div}$ increase, the model consistently improves performance compared to the value set as 0. It confirms the effectiveness of our losses. In Figure 7(a), the performance rises consistently on two datasets, which verifies that fine-grained interactions are essential for knowledge transfer. In Figure 7(b-c), we see a uniform performance improvement in AWA2 with the increase of $\lambda_{var}$ and $\lambda_{div}$. It verifies that $\mathcal{L}_{var}$ and $\mathcal{L}_{div}$ aid the model to generate multi-view semantic embeddings with information decoupling, facilitating the partial semantic alignment. The results in FLO demonstrate the same conclusion when $\lambda_{var} \geq 0.5$ and $\lambda_{div} \geq 1.0$.

**Effect of Number of View Embeddings $k$ in SDM.** In Figure 7(d), we report the H when varying $k$ from 1 to 8. Besides, we compare with $k = 0$, the baseline without SDM, *i.e.*, leveraging global feature as the view embedding. As the $k$ increases, we see progressive performance improvement across both datasets. This is due to multi-view embeddings capturing distinct semantics. However, a high $k$ may be biased to seen classes, thus harming the performance.

## 4.6 Computation Cost Analysis

In Table 5, we compare the trainable parameters, the time for training one epoch and inference single image, and the performance with previous methods. With comparable computation cost, our method outperforms previous methods [37, 38]. It verifies that the performance improvement is due to more accurate semantic alignment instead of increased parameters. Moreover, after adding the SDM, the performance improves significantly with a slight increase in

**Table 5: Computation cost analysis on the FLO dataset.**

| Model | Params ($\times 10^6$) | Train (min) | Inference (ms) | FLO (H) |
|---|---|---|---|---|
| I2DFormer [38] | 2.18 | 0.72 | 4.7 | 53.8 |
| I2MVFormer [37] | **3.86** | 0.80 | **5.3** | 57.1 |
| **EmDepart** w/o SDM | 1.52 | 0.67 | 4.6 | 57.9 |
| **EmDepart** | 3.10 | **0.98** | 5.2 | **67.3** |

model parameters and training time. This indicates the importance of decomposing semantics for modeling the partial association.

## 5 Conclusion

Our EmDepart models the partial association between documents and corresponding images, accurately aligning visual and textual space based on the matching information. By introducing local-to-semantic variance loss and multiple semantic diversity loss, SDM generates multi-view semantic embeddings. These losses also help the previous methods solve the feature collapse problem. Moreover, we introduce two losses to partially align the semantic concepts between documents and images at the view and word-to-patch levels. In addition, we propose a partial score to filter out unmatched information and evaluate semantic similarity accurately. With comparable training parameters, EmDepart outperforms SOTA methods on three benchmarks for document-based ZSL. Qualitatively, our model learns the interpretable partial semantic association.

## Acknowledgments

This work was supported by the Central Guidance for Local Special Project (Grant No. Z231100005923044) and the Climbing Plan Project (Grant No. E3Z0261).

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
