# OpenReview forum: "Visual-Semantic Decomposition and Partial Alignment for Document-based Zero-Shot Learning"
_acmmm.org/ACMMM/2024/Conference — MM2024 Poster_

### Official Review · Reviewer_3YLN · 2024-05-22

**Rating:** 4
**Confidence:** 3

**Summary:**

This paper proposes a method called EmDepart (Embedding Decomposition and Partial Alignment) for document-based zero-shot learning (ZSL). The proposed method introduces the Semantic Decomposition Module (SDM) to generate multi-view semantic embeddings and uses the local-to-semantic variance loss and multiple semantic diversity loss to capture unique local details and enhance orthogonality among the embeddings. It also applies partial semantic alignment at the view and word-to-patch levels. The experimental results show that EmDepart outperforms state-of-the-art methods in document-based ZSL on three standard benchmarks.

**Strengths:**

1. EmDepart proposes an innovative approach to solve the problem of partial document-image correlation, introducing the concepts of multi-view semantic embedding and partial alignment, which improves the effect of zero-shot learning.
2. The paper is well-organized and easy to read.

**Limitations:**

1. The semantic decomposition module relies on perceived features from different modalities, such as shape, color, and habitat.  However, this approach may not capture all the nuances and complexities of the semantic concepts, leading to a limited representation of the semantics.
2. It does not consider the partial association between noisy documents and visually diverse images. This can lead to suboptimal alignment, especially when dealing with non-visual views in documents or diverse visual content in images.

**Suitability:**

3

---

### Official Review · Reviewer_K6CV · 2024-05-24

**Rating:** 4
**Confidence:** 3

**Summary:**

This paper proposes a novel network that partially aligns textual and visual space according to their semantic-relevant degree.
This addresses the suboptimal alignment caused by ignoring the incomplete semantic equivalence between documents and images.

**Strengths:**

1. The Semantic Decomposition decomposes features into multi-view semantic embeddings, facilitating the partial alignment of matching semantics.  And two losses are introduced to enhance distinctiveness and orthogonality among embeddings, addressing the issue of feature collapse and improving previous methods.
2. State-of-the-art results are achieved on three benchmarks, demonstrating the effectiveness of the approach.
3. The model learns interpretable partial semantic associations, which are visualized effectively.

**Limitations:**

1. Limited Comparative Analysis: The paper could benefit from more comparative experiments, such as comparisons with more ZSL methods, to further validate the approach.
2. Robustness Analysis: Evaluating the model's robustness could provide a more comprehensive understanding of its real-world applicability, and a sensitivity analysis on the number of view embeddings k could help in understanding model stability.

**Suitability:**

2

---

### Official Review · Reviewer_UMVS · 2024-05-25

**Rating:** 4
**Confidence:** 3

**Summary:**

This paper proposes a novel network, termed Embedding Decomposition and Partial Alignment (EmDepart), that partially aligns textual and visual space according to their semantic-relevant degree for Zero-Shot Learning (ZSL). Specifically, the Semantic Decomposition Module (SDM) is proposed to generate multi-view semantic embeddings from visual and textual sides, providing the basic concepts for partial alignment. And the local-to-semantic variance loss is to capture unique local details and multiple semantic diversity loss is to make each embedding orthogonal to others. Moreover, a novel score is applied to filter out unmatched information to measure semantic similarity accurately at the inference. The experimental results on three benchmark datasets show its effectiveness.

**Strengths:**

1. This paper is well-written and easy to follow.
2. The paper focuses on the limitations of global alignment between documents and images, which has practical research value, and the method proposed is also relatively novel.
3. The proposed method is effective, and sufficient experiments have been conducted for analysis.

**Limitations:**

1. There are some details errors, such as some abbreviations (CLSN, IMG) not being mentioned.
2. Local alignment between vision and semantics has been studied in some works using attributes as auxiliary information (such as Transzero [1], DAZLE [2] and Gndan[3]), but the related research has not been comparatively discussed.


> - [1] Chen S, Hong Z, Liu Y, et al. Transzero: Attribute-guided transformer for zero-shot learning[C]//Proceedings of the AAAI conference on artificial intelligence. 2022, 36(1): 330-338.
> - [2] Huynh D, Elhamifar E. Fine-grained generalized zero-shot learning via dense attribute-based attention[C]//Proceedings of the IEEE/CVF conference on computer vision and pattern recognition. 2020: 4483-4493.
> - [3] Chen S, Hong Z, Xie G, et al. Gndan: Graph navigated dual attention network for zero-shot learning[J]. IEEE transactions on neural networks and learning systems, 2022.

**Suitability:**

2

---

### Official Review · Reviewer_2MkN · 2024-05-30

**Rating:** 5
**Confidence:** 3

**Summary:**

This paper focus on the problem of partial association between noisy documents and visual-diverse images in zero-shot learning (ZSL). They propose an Embedding Decomposition and Partial Alignment (EmDepart) network to extract multi-view semantic concepts from document and image and accurately align the matching concepts. Furthermore, to solve the issue of information redundancy caused by feature collapse, the authors introduce the semantic decomposition module with the local-to-semantic variance loss and multiple semantic diversity loss.

**Strengths:**

- This paper is well-written and easy to following.
- The research problem is meaningful and the proposed solution is reasonable.
- The proposed method has a strong performance across common ZSL datasets.
- Extensive experiments further illustrate the effectiveness of the proposed method.

**Limitations:**

- Could the Semantic Decomposition Module (SDM) be adapted to attributes-based ZSL methods?

**Suitability:**

3

---

### Meta-Review · Area_Chair_mcBF · 2024-07-03

**Recommendation:** Accept (Poster)
**Confidence:** 5

**Metareview:**

This paper proposes a novel network to partially align textual and visual space according to their semantic-relevant degree for Zero-Shot Learning (ZSL). All reviewers unanimously recommend accepting the paper as the rebuttal has properly addressed their concerns.

---

### Meta-Review · Senior_Area_Chairs · 2024-07-10

**Recommendation:** Accept (Poster)
**Confidence:** 4

**Metareview:**

All the reviewers gave positive ratings and tend to accept the paper. SAC and AC agree with reviewers and recommend acceptance of the paper.